# Quality of life in adults with Down syndrome: A mixed methods systematic review

**Ogochukwu Ann Ijezie**[1]*, **Jane Healy**[2], **Philip Davies**[1], **Emili Balaguer-Ballester**[1,3],
**Vanessa Heaslip**[4,5]

**1** Department of Computing and Informatics, Bournemouth University, Poole, United Kingdom,
**2** Department of Social Science and Social Work, Bournemouth University, Lansdowne, United Kingdom,
**3** Bernstein Centre for Computational Neuroscience, Medical Faculty Mannheim, Heidelberg University,
Mannheim, Germany, **4** School of Health and Society, University of Salford, Manchester, United Kingdom,
**5** Department of Social Studies, University of Stavanger, Stavanger, Norway

\* oijezie@bournemouth.ac.uk

## Abstract

### Background

As the life expectancy of adults (aged $\geq$ 18 years) with Down syndrome increases for a
plethora of reasons including recognition of rights, access, and technological and medical
advances, there is a need to collate evidence about their quality of life.

### Objective

Using Schalock and Verdugo's multidimensional quality of life assessment model, this sys-
tematic review aimed to identify, synthesise and integrate the quantitative and qualitative
evidence on quality of life in adults with Down syndrome via self-and proxy-reporting.

### Methods

Five databases were systematically searched: MEDLINE, CINAHL, PsycINFO, Scopus,
and Web of Science to identify relevant articles published between 1980 and 2022 along
with grey literature and reference lists from relevant studies. A mixed methods systematic
review was performed according to the Joanna Briggs Institute methodology using the con-
vergent integrated approach. The review followed the Preferred Reporting Items for Sys-
tematic Reviews and Meta-Analyses guidelines.

### Results

Thirty-nine studies were included: 20 quantitative, 17 qualitative, and 2 mixed methods
studies. The synthesised findings were grouped into the 8 core domains of quality of life:
personal development, self-determination, interpersonal relations, social inclusion, rights,
emotional, physical and material well-being. Of the 39 studies, 30 (76.92%) reported on
emotional well-being and 10 (25.64%) on rights. Only 7 (17.94%) studies reported that
adults with Down syndrome have a good quality of life centred around self-determination
and interpersonal relations. Most adults with Down syndrome wanted to become more inde-
pendent, have relationships, participate in the community, and exercise their human rights.

University - Malaysia Campus: Xiamen University -
Malaysia, MALAYSIA

**Data Availability Statement:** All relevant data are
within the paper and its Supporting information
files.

**Funding:** The authors received no specific funding for this work.

**Competing interests:** The authors have declared that no competing interests exist.

Self-reported quality of life from adults with Down syndrome was rated higher than proxy reported quality of life. Discrepancies in quality of life instruments were discovered.

## Conclusion

This review highlighted the need for a better systematic approach to improving the quality of life in adults with Down syndrome in targeted areas. Future research is required to evaluate self-and proxy-reporting methods and culture-specific quality of life instruments that are more appropriate for adults with Down syndrome. In addition, further studies should consider including digital assistive technologies to obtain self-reported quality of life data in adults with Down syndrome.

## International prospective register of systematic reviews registration number

CRD42019140056.

## Introduction

Down syndrome (DS) is the most common genetic cause of intellectual and developmental disability (IDD) resulting from the presence of an extra copy of chromosome 21 [1, 2]. Globally, there are approximately 1:1000 to 1:1100 live births of people with DS [3]. Epidemiological evidence suggests that the average life expectancy of people with DS is now over 60 years [4]. As adults with DS live longer, healthcare professionals have more opportunities to understand these individuals' needs [5]. As defined by the World Health Organisation (WHO), quality of life (QoL) is an "individual's perception of their position in life in the context of the culture and value systems in which they live and in relation to their goals, expectations, standards and concerns" ([6], p.1405). Several studies suggest the importance of using subjective and objective measures in providing a holistic QoL assessment for people with DS [7, 8]. Nevertheless, obtaining self-reported QoL data from this group has proven difficult due to speech impediments [9], cognitive impairments [10, 11], response biases [12], and challenges in obtaining informed consent [13].

Digital assistive technologies such as augmentative and alternative communication (AAC) play a crucial role in supporting adults with IDD such as DS who have language and communication difficulties, promoting self-determination and social participation [14–16]. The integration of speech recognition technology into AAC for adults with DS, such as using dictation for word processing or text messaging and using smart speakers for reminders can improve their QoL [17]. In the population of adults with DS, the use of machine learning or artificial intelligence techniques such as natural language processing in speech recognition software could lead to exponential growth in supporting complex communication needs [18, 19]. Most adults with IDD currently use technological devices such as smartphones [20–22], tablets [23], desktops or laptops [24] and assistive products [25]. A systematic review conducted by Krasniqi and colleagues [26] highlighted the importance of digital assistive technology to support the process of acquiring skills needed to solve real-world problems for adults with DS. Proponents of the social model of disability hold opposing perspectives regarding the risk of the oppressive and disabling nature of digital assistive technologies for people with IDD on all forms of abuse, including online abuse, name-calling, sexual victimisation, and extortion affecting their QoL

[27, 28]. There is evidence that probable barriers to digital assistive technology could be caused by a lack of funding and device design issues [29].

While consensus about the role of digital assistive technologies is still evolving, a growing body of literature recognises the importance of researching the QoL of people with DS, in terms of its contribution to assessing personal outcomes and guiding organisational and system-level policies to improve lives [30]. Therefore, it is important to identify factors that might improve or decrease QoL scores beyond childhood and adolescence, including the transition to and during adulthood. Empirical studies show that overall QoL is lower in children with DS than in children without DS, albeit at different levels across the QoL domains [31, 32]. Children with DS show moderate or favourable levels of QoL in most domains except emotional well-being [33]. In contrast, little attention has been paid to whether this applies to adults with DS. Evidence from existing studies encourages self-reported QoL from people with DS and should be considered a priority for assessing their QoL [34–36]. There are recognised methods for self-reporting QoL by people with DS, such as reliable and valid, easy-to-understand measurement scales [36, 37], semi-structured interviews [38], and image-based methods [39]. Assessing the QoL of people with IDD requires self- and proxy-assessment strategies to check for consistency and differences in both reports [40, 41] such as DS. To date, there are conflicting reports in the literature on the inter-rater reliability and concordance of the two assessment strategies: self-report and proxy report [42, 43]. Inter-rater reliability refers to the overall agreement between different raters [44]. Existing research indicates that when individuals are unable to self-report, their family members or caregivers act as proxies to provide additional information [45, 46]. However, it is argued that proxy ratings cannot accurately reflect the QoL of people with DS [42, 43].

Although previous scoping reviews have examined QoL in children and adolescents with DS [42, 47], a systematic review on the assessment of QoL in adults with DS has not yet been conducted. The Schalock and Verdugo conceptual model on QoL has been widely accepted, critically evaluated, validated in different cultures, and used to assess the QoL of people with IDD [48–50]. The model consists of eight domains: 1) personal development (e.g., education, personal competence); 2) self-determination (e.g., autonomy, choices); 3) interpersonal relations (e.g., interactions, relationships); 4) social inclusion (e.g., community integration, social supports); 5) rights (e.g., human, legal); 6) emotional well-being (e.g., contentment, self-concept); 7) physical well-being (e.g., health, leisure); 8) material well-being (e.g., financial status, employment). This systematic review aims to identify, synthesise and integrate the quantitative and qualitative evidence on the QoL in adults (aged $\geq$ 18 years) with DS via self-reports and proxy reports using Schalock and Verdugo's QoL model. The review includes studies reporting QoL outcomes in adults with DS. This broad focus allows for a comprehensive review of existing evidence on the QoL in this population. In addition, this review provides good practice recommendations for advancing QoL research on DS.

## Materials and methods

### Study design

A mixed methods systematic review was conducted following the Joanna Briggs Institute (JBI) methodology using the convergent integrated approach [51] to answer the review aim. Mixed methods systematic reviews combine quantitative and qualitative evidence to create a breadth and depth of understanding of the phenomenon of interest and to inform evidence-based practice [51, 52]. Based on the typology of systematic reviews developed by Hong et al. [53] and the work by Sandelowski and colleagues [54, 55], the JBI methodology was developed [51]. This review was undertaken in accordance with the Preferred Reporting Items for

Systematic Reviews and Meta-analyses (PRISMA) guidelines (S1 File) [56]. The study protocol has been registered with the International Prospective Register of Systematic Reviews (PROSPERO), registration number: CRD42019140056.

## Search strategy

The keywords required for the search were identified using the Population, Exposure and Outcome (PEO) framework [57] to guide the search and obtain the specific studies appropriate for review. A structured search strategy was developed in consultation with two experienced librarians. A systematic literature search was conducted in five bibliographic databases: MEDLINE, CINAHL, PsycINFO, Scopus, and Web of Science to identify relevant articles published from 1980 to 2022. QoL was first used in IDD in 1980 [58]; therefore, studies from this time point were selected. An iterative process using controlled vocabulary, synonyms, related terms and subject headings interconnected by Boolean operators ("AND" and "OR" only) was employed for the query search development. The search was conducted using the keyword combinations: *Down\* syndrome*, *Trisomy 21*, *Quality of Life*, and *Well-being* as detailed in S1 Table. Reference lists of relevant publications and grey literature were hand-searched to identify additional studies not identified in the initial electronic search for an exhaustive search process.

## Eligibility criteria

According to the PEO framework [57], the eligibility criteria are listed in Table 1.

## Study selection

Titles, abstracts and the full-text articles of potentially relevant studies were manually screened by the first author according to the eligibility criteria. As part of quality assurance, the co-

**Table 1. Eligibility criteria.**

| | Inclusion criteria | Exclusion criteria |
|---|---|---|
| **Population (P)** | • Studies involving human participants who are adults aged ≥ 18 years. | • Studies in people < 18 years of age, studies that combined three age groups: children, adolescents, and adults, but did not but did not analyse the data explicitly for each group. |
| **Exposure (E)** | • Studies on DS and Trisomy 21. | • Studies of other chromosomal disorders or other IDDs with no known cause and adults with DS with dementia. |
| **Outcome (O)** | • Studies on QoL and well-being of adults with DS using self-report and/or proxy reports.<br>• Studies that combined children, family members/caregivers and adults with DS, but QoL data from adults with DS were collected and analysed separately.<br>• Studies that combined IDD and DS but explicitly collected and analysed QoL data for adults with DS separately. | • Studies on FQoL, HRQoL and studies that combined IDD and DS but did not separately analyse and report QoL data from adults with DS. |
| **Study design** | • Quantitative (cross-sectional and cohort or longitudinal studies), qualitative (interviews, focus groups, case studies, image-based methods) and mixed methods studies. | • Clinical trials evaluating the potency of a drug or medical intervention in relation to a clinical outcome. |
| **Other** | • Studies in English language only. | • Studies published in languages other than English due to a lack of resources for translating data.<br>• Commentaries, abstracts only, conference proceedings, consensus statements, reviews, case reports, case series, dissertations, and articles on ethical or legal issues. |

FQoL = Family quality of life; HRQoL = Health-related quality of life.

authors checked (n = 40; 24.24%) potentially eligible full-text articles. Any disagreements were resolved through discussion and a final decision was made using a consensus-based approach.

## Data extraction and quality appraisal

Data extraction and quality appraisal of studies were performed in Microsoft Excel 365 (Microsoft Inc., Redmond, WA) by the first author and cross-checked by co-authors to ensure accuracy. For the data extraction, the following characteristics were extracted and tabulated: study information (reference and geographical location of study), study design, participant characteristics, QoL domains, QoL measures, QoL assessment report methods and key findings (S2 Table). Due to the heterogeneity of the study designs, the Quality Assessment for Diverse Studies (QuADS) [59] was used to appraise the studies. This tool is a revised version of the Quality Assessment Tool for Studies with Diverse Designs (QATSDD) [60]. The QuADS tool shows strong inter-rater reliability and content and face validity [59]. It has a 13-item tool for evaluating studies on a 3-point scale from 0 to 3. None of the studies was excluded based on the quality appraisal as they provided valuable information, leading to a comprehensive review.

## Data synthesis

In accordance with the JBI methodology, a convergent integrated approach was applied [51, 52] which involved 'qualitising' the quantitative data (via data transformation) into textual descriptions to allow integration with the extracted qualitative data to answer the review aim (S2 Table). For the mixed methods studies, each strand was also analysed independently. Qualitised data were assembled and then pooled with the extracted qualitative data to identify categories based on similarity in meaning, to produce the overall integrated findings of the review and draw up recommendations to inform evidence-based policy. These categories were mapped to the eight core QoL domains of the conceptual framework of Schalock and Verdugo [58]. This process was conducted by the first author and was discussed and agreed upon with co-authors.

# Results

## Literature search results

A total of 1,466 articles were identified through database searches and exported to EndNote 20.0.1 reference management software (Clarivate Analytics, Philadelphia, PA), of which 570 were duplicates. A total of 731 articles were excluded during the title and abstract screening phase. Of the remaining 165 articles, 139 articles were excluded based on inclusion and exclusion criteria during full-text check. Additional searches of grey literature such as ResearchGate, ProQuest, and reference lists of all relevant articles were performed for a complete list, and 13 additional articles were identified. A total of 39 items were included in the review (Fig 1).

## Description of included studies

The results section first describes the study characteristics and details how QoL was assessed in this population. Of the 39 included studies, 18 (46.15%) were quantitative cross-sectional, 2 (5.12%) were quantitative longitudinal, 17 (43.58%) were qualitative and 2 (5.12%) were mixed methods (S2 Table). The sample sizes ranged from 5 to 1857 participants with male and female adults with DS and different age groups (18 to 60 years). Based on the World Bank Classification on country classification [61], all studies were conducted in high income countries (HICs), with most emerging from the United States of America, as shown in Fig 2.

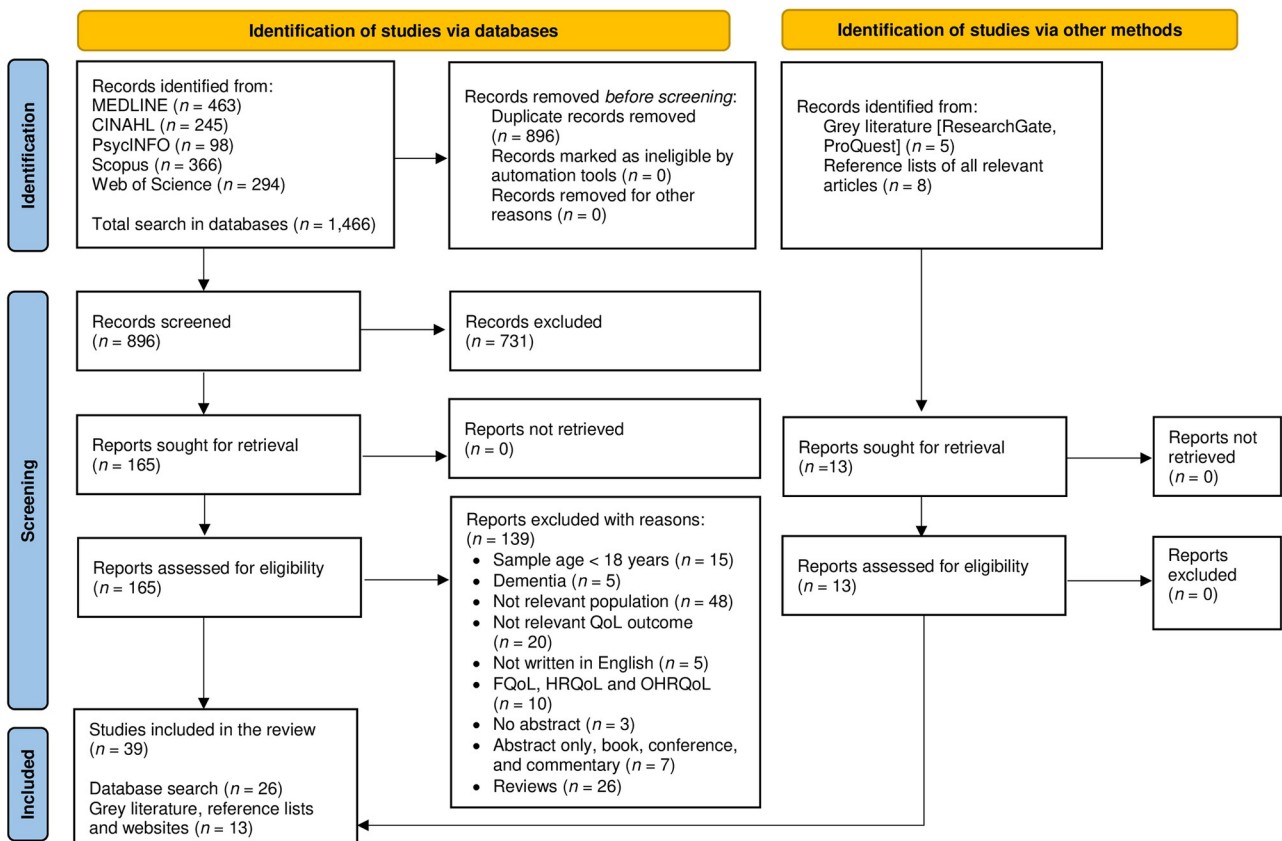

**Fig 1. PRISMA flowchart of the study selection and inclusion process. Note:** QoL = Quality of life; OHRQoL = Oral health-related quality of life.

No studies in low-and middle-income countries (LMICs) were identified after the PRISMA process.

Eight studies collected self-reported QoL data directly from adults with DS [62–69]. Four studies obtained self-reports with minimal support from the proxies [70–73]. Eleven studies collected proxy reported QoL from caregivers [74–84] and 16 from a combination of self and proxy reports [85–100].

## Instruments used to assess QoL

In the cross-sectional and longitudinal studies, QoL was assessed using 51 different instruments, of which 47 were QoL instruments and battery of tests and four were DS-specific QoL instruments (Table 2). Of the DS-specific QoL instruments, two were modified scales to measure the QoL in adults with DS and the remaining two measured only emotional well-being domain.

## Quality appraisal findings

The included studies varied in their scores ranging from 16/39 (41.03%) to 35/39 (89.74%) using the QuADS tool (S3 Table). High scores were reported in the statement of research aims, description of research setting and target population, study design appropriate to address the research aim, the format and content of data collection tool provided, recruitment data provided, and the method of analysis. Low scores were reported in terms of appropriate sampling

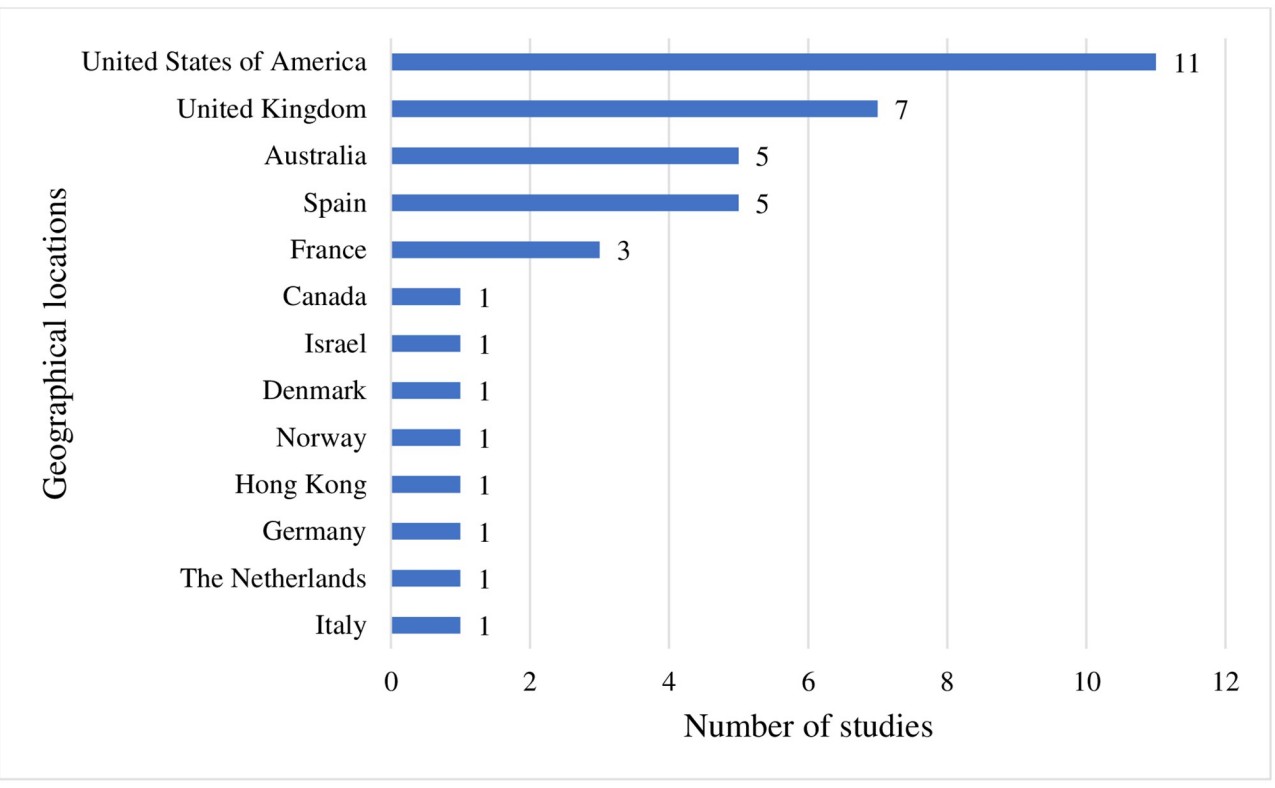

**Fig 2. Distribution of included studies across countries.**

to address the research aim, theoretical or conceptual underpinning to the research, rationale for choice of data collection tool, procedure in recruitment data, justification for analytic method selected, evidence that stakeholders had been considered in the research design or conduct and critical discussion of strengths and limitations.

## Findings by QoL domain

The QoL measures and indicators contained in the extracted data were classified into the eight QoL domains of Schalock and Verdugo and presented as follows: personal development, self-determination, interpersonal relations, social inclusion, rights, emotional well-being, physical well-being, and material well-being [58]. An additional section entitled "Overall QoL and coverage of the QoL domains" was added that summarises studies encompassing all eight QoL domains reported across studies. Studies that included proxy reports were obtained from different sources: family members (fathers and/or mothers, siblings, or other relatives), educators, sponsors, workplace supervisors, carers/caregivers, special schoolteachers, professionals, and staff.

## Personal development

A total of 24 studies focused on the personal development of adults with DS. Four self-reported [62, 64, 65, 68], two parent-proxy reports [74, 76] and four studies using both self-and proxy-reported [90, 93, 95, 100] examined the educational status of adults with DS who had attended mainstream schools and further education colleges. Two studies [76, 90] reported that adults

**Table 2. QoL instruments (from quantitative studies) identified in the included studies.**

| Acronyms | Complete label | Number of items | Number of dimensions | Studies which used instrument in this review |
|---|---|---|---|---|
| **Generic QoL instruments and battery of tests** | | | | |
| CDI-S | Children's Depression Inventory Short Form | 27 | 1 | Ailey et al. (2006); Heller et al. (2004) [87, 92] |
| PIMRA-AD | Psychopathology Instrument for Mentally Retarded Adults Affective Disorders Subscale | NA | 1 | Ailey et al. (2006) [87] |
| PSSQ | Perceived Social Support Questionnaire | 4 | 1 | Ailey et al. (2006) [87] |
| LQ | Loneliness Questionnaire | 6 | 1 | Ailey et al. (2006) [87] |
| LSS | Life Satisfaction Scale | 16 | 5 | Ailey et al. (2006); Heller et al. (2004) [87, 92] |
| Brief WAIS test | Brief version of Wechsler Adult Intelligence Scale test | NA | NA | Brown (1994) [88] |
| SEAFI | Social Education Adaptive Functioning Index | NA | NA | Brown (1994) [88] |
| VAFI | Vocational Adaptive Functioning Index | NA | NA | Brown (1994) [88] |
| SAMU-DISFIT | Servicios de Asistencia Médica de Urgencias Disability Fitness Battery[a] | 8 | 4 | Cabeza-Ruiz et al. (2019) [75] |
| – | Quality of life questionnaire | NA | 10 | Brown (1994) [88] |
| MI scale | Malaise Inventory Scale and questions on leisure interests and experiences by Holmes [101] | NA | NA | Carr (2008) [76] |
| ABS | Adaptive Behaviour Scale | NA | 10 | Collacott (1992) [77] |
| – | Combination of Instrumental Activities of Daily Activities and Activities of Daily Living scales[b] | 15 | NA | Heller et al. (2004) [92] |
| C-EBES | Cognitive–Emotional Barriers to Exercise Scale | 9 | NA | Heller et al. (2004) [92] |
| EPS | Exercise Perceptions Scale | 9 | NA | Heller et al. (2004) [92] |
| SS | Self-Efficacy to Exercise Regularly scale of the Lorig Self-Efficacy to Perform Self-Management Behaviours Instrument | 5 | NA | Heller et al. (2004) [92] |
| CIS | Community Integration Scale | NA | NA | Heller et al. (2004) [92] |
| SUS | System Usability Scale | 10 | 3 | Landuran and N'Kaoua (2021); Landuran et al. (2022) [73, 85] |
| QUEST 2.0 | Quebec User Evaluation of Satisfaction | NA | NA | Landuran and N'Kaoua (2021); Landuran et al. (2022) [73, 85] |
| RSES | Rosenberg Self-Esteem Scale | NA | NA | Landuran and N'Kaoua (2021); Landuran et al. (2022) [73, 85] |
| PWB | The Psychological well-being scales | NA | 6 | Landuran and N'Kaoua (2021); Landuran et al. (2022) [73, 85] |
| PSWQ | The Penn State Worry Questionnaire | NA | NA | Landuran and N'Kaoua (2021); Landuran et al. (2022) [73, 85] |
| EPADV | The scale of perceived self-determination in the domains of life | NA | NA | Landuran and N'Kaoua (2021); Landuran et al. (2022) [73, 85] |
| ARC | Association for the Retarded Citizens self-determination scale | 72 | 4 | Landuran and N'Kaoua (2021); Landuran et al. (2022) [73, 85] |
| WISC-IV | Brief version of the Wechsler Intelligence Scale for Children IV | NA | 4 | Landuran et al. (2022) [66] |
| CBT | Corsi Block Test[a] | NA | NA | Landuran et al. (2022) [66] |
| RBMTC | Version A of the Rivermead Behavioural Memory Test for Children[a] | NA | NA | Landuran et al. (2022) [66] |
| 5TSTS | The Five Times Sit to Stand Test[a] | NA | NA | Landuran et al. (2022) [66] |
| TUG test | The Timed Up and Go Test[a] | NA | NA | Landuran et al. (2022) [66] |
| SPPB | Short Physical Performance Battery[a] | NA | NA | Landuran et al. (2022) [66] |
| PEBL version 0.14 | The Psychology Experiment Building Language version 0.14[a] | NA | NA | Landuran et al. (2022) [66] |
| IALS | The Inventory of Apartment Living Skills | NA | NA | Landuran et al. (2022) [73] |
| WHOQOL | World Health Organisation Quality of Life—Bref | 26 | 4 | Landuran et al. (2022) [73] |
| SB5 | The Stanford-Binet, Fifth Edition Abbreviated Battery[a] | NA | NA | Mihaila et al. (2017) [82] |

*(Continued)*

**Table 2.** (Continued)

| Acronyms | Complete label | Number of items | Number of dimensions | Studies which used instrument in this review |
|---|---|---|---|---|
| RSMB | The Reiss Screen for Maladaptive Behaviour | 26 | NA | Mihaila et al. (2017) [82] |
| VLS | Victoria Longitudinal Study activity questionnaire | 68 | 10 | Mihaila et al. (2017); Mihaila et al. (2020) [82, 86] |
| TLAB | Trail Leisure Assessment Battery for People with Cognitive Impairments[a] | 10 | NA | Mihaila et al. (2020) [86] |
| – | Combination of Modified Vineland Social Maturity Scale and Denver Development Scale [b] | 83 | 8 | Schroeder-Kurth et al. (1990) [84] |
| InterRAI-ID | International Resident Assessment Instrument Intellectual Disability | NA | 4 | Villani et al. (2020) [99] |
| DRS | Depression Rating Scale | NA | NA | Villani et al. (2020) [99] |
| ABS | Aggressive Behavior Scale | NA | NA | Villani et al. (2020) [99] |
| SOCWD | Social Withdrawal Scale | NA | NA | Villani et al. (2020) [99] |
| COMM | Communication Scale | NA | NA | Villani et al. (2020) [99] |
| CPS | Cognitive Performance Scale | NA | NA | Villani et al. (2020) [99] |
| ADLH | Activities of Daily Living Hierarchy | NA | NA | Villani et al. (2020) [99] |
| IADLH | Instrumental Activities of Daily Living Hierarchy | NA | NA | Villani et al. (2020) [99] |
| PS | Pain scale | NA | NA | Villani et al. (2020) [99] |
| **DS specific QoL instrument** | | | | |
| Modified KidsLife-Down scale | A modified version of the KidsLife-Down scale | NA | 8 | Camacho et al. (2021) [89] |
| EQ-i: SVDS | Emotional Quotient Inventory: Short Version for Down Syndrome | 25 | 4 | Sánchez-Teruel et al. (2020) [72] |
| EQ-i:YV | Bar-On Emotional Quotient Inventory: Youth Version | 60 | 5 | Robles-Bello et al. (2022) [69] |
| – | A modified version of a Spanish Quality of life questionnaire[c] | NA | 4 | Pérez et al. (2018) [71] |

Note:

[a] Battery of tests as reported in the included studies;

[b] The authors of the included studies combined the scales into a single scale;

[c] The questionnaire was specifically developed for adults with Down syndrome, NA = Not available.

with DS were in their late 30s at the time of the study and in school, with 79% reading and/or look through books at least once a week [76]. Parents, especially mothers, played an important role in adults with DS in encouraging them to continue their education. For example, four studies reported that parents advocated for the need for educational services and programs for adults with DS [62, 74, 78, 100]. Three studies reported that family and school support are important factors for self-development through active participation in extracurricular activities [95, 98, 100]. One study [70] reported that compared to other disability groups, adults with DS had the lowest support needs for behavioural problems and another study [97] showed that their adjustment behaviours were also below expectations based on chronological age, and all showed significant cognitive and academic deficits in standardised ratings. A study using both self-reports and proxy reports found that adults with DS reported higher self-perceptions of personal development than proxy reports, which reported lower scores [89].

Ten studies [62, 64, 65, 68, 82, 84, 86, 88, 95, 100] found that adults with DS demonstrated reading and writing skills, four studies [62, 68, 84, 88] reported numeracy skills, and four studies [62, 64, 94, 95] reported on computer skills. Female adults with DS showed a significant difference in literacy and numeracy skills compared to males, while male adults with DS scored slightly higher in concept attainment and money skills [88]. While another study reported that male adults with DS showed higher adaptive behaviour, competence, social skills, and better

communication development compared to female adults with DS [89]. Two studies reported that adults with DS could not handle money, measure weight, had poor shopping skills and were uncertain about winning or losing [77, 90], although individuals aged 18 to 49 years performed better [77]. Adults with DS expressed awareness of communication difficulties, stuttering, not being heard, difficulties with communication partner [93], a significant reduction in comprehension in those aged 50–59 years, a deterioration in social language and expressive language after 60 years [77] and found no significant changes in communication problems before and after lockdown during the coronavirus disease 2019 (COVID-19) pandemic [99]. Two studies reported on the design of a life plan via information and communication technology (ICT) for adults with DS based on their responses to make projections and future plan [73, 85].

## Self-determination

Twenty-five studies examined self-determination in adults with DS. In the self-determination QoL domain, some comparisons between adults with DS and some groups. For example, one study examined three different types of disability, adults with DS were ranked second compared to other disability groups to account for most of the decision variables, i.e., choice of a daily routine, people to live with, their case manager, and what to buy with their allowance [70]. In another study involving three groups of adults with DS and a control group of adults, the more self-determined DS groups had better memory, motor and language skills than the less self-determined individuals compared to a control group [66]. High scores were obtained in more self-determined DS groups and control groups of adults for memory, motor skills and language skills, and it was emphasised that the intelligence quotient (IQ) of the three DS groups did not differ significantly and was not correlated with self-determination [66]. Similarly, another study reported an improvement in self-determination related to autonomy and home skills in the experimental group compared to the control group of adults with DS [73]. A proxy reported study found that adults with DS living in institutions needed help with daily tasks, which was a contrast to those living in families; in both groups, approximately 50% were fully independent in personal care [84]. A study that used both self- and proxy-reports found statistical differences in self-determination as proxies reported that adults with DS had difficulty participating in their environment independently and to make autonomous decisions [89].

Three studies reported that the use of ICT by adults with DS resulted in a significant increase in their well-being, which was associated with autonomy and an improvement in perceptions of self-determination in everyday life, e.g., daily tasks, contact with family and friends, using useful apps, phone calendar, social media and setting up reminders [64, 73, 85]. Thirteen studies showed that the freedom to have their own rules, to live independently, and move out of home without the influence of their parents were paramount and considered a symbol of adulthood to adults with DS [62–64, 68, 74, 76, 88, 90, 91, 93, 95, 97, 98]. Five studies reported that some adults with DS demonstrated their ability to live independently such as catching a bus [93], taking responsibility for personal hygiene, house cleaning, laundry, cooking, paying bills, budgeting, taking medication and using public transport [63, 64, 97] and making decisions [65]. A proxy reported study found that young adults with DS tend to find difficult to attain desired social roles as adults [78].

Eight studies documented that regardless of the independence of adults with DS, some required support in going for walks, going to the cinema, staying healthy, seeing a doctor [63, 98], emotional support [63], participation in extracurricular activities [95], personal care [74, 100], handling money [88, 100], respite care, using public transport [78, 100] and want to

continue living with parents [64]. In three studies, most adults with DS became more independent as they transitioned to adulthood and caregivers recognised the importance to their adults with DS QoL [91, 96, 98]; while one study documented that most mothers did not remember whether young adults with DS played an integral part of the transition process or were actively involved in decision making [78].

Seven studies found that most mothers were concerned about their adults with DS, in terms of having to live independently, being responsible for their decisions [74, 83, 91, 98], being vulnerable to sexual abuse/exploitation in sexual relationships [79, 80], leaving the house unsupervised at night [79], and being left alone [76]. Two studies found caregivers (mainly mothers) wanted their adults with DS to be independent and make daily decisions [78, 95] and seek supported accommodation, but it was unlikely to be achieved [78]. A proxy reported study found that mothers advised themselves to advocate for adults with DS when they turn 18 on getting legal advice and gaining independence [74].

## Interpersonal relations

Twenty studies examined interpersonal relations in adults with DS. Of these, 11 studies reported that the greatest support for adults with DS was their family, who provided them with a safe place to live, encouraged, and advocated for them [62–64, 68, 79, 80, 83, 91, 95, 97, 98]. Three studies reported that caregivers, particularly, mothers [78] and family members [91, 98] were strongly involved in the adulthood transition process. Despite valuing parental guidance, a desire for independence and recognition as an adult with DS often led to disagreements where caregivers were perceived as too controlling or imposing too many rules [64, 68, 98]. Six studies reported that caregivers treated the adults with DS as they would treat non-disabled individuals by helping them and offering them opportunities to participate in various activities [74, 79, 80, 83, 95, 98] and created opportunities for them rather than focusing on limitations [98]. A study reported that family members, friends, and guardians visited and contacted adults with DS living in institutions or sheltered housing, but some were not visited or contacted [81].

Seven studies reported that adults with DS were passionate about their friendships and rated them as a major contributor to their sense of social inclusion, acceptance, and self-esteem [62, 63, 68, 91, 95, 96, 98]. However, some studies reported that adults with DS had few friends [63, 93], 40% wanted more friends or had no friends [78, 87], whereas other studies found adults with DS had many friends [98] or made friends easily [76]. Although some mothers mentioned their adults with DS made friends from their siblings' friends [79]. One study using both self- and proxy reports found that adults with DS were more likely to nominate friends and colleagues on their social networks rather than proxies (family members), while proxies were more likely to nominate paid staff and organisations [96]. In the same study, network members such as family, friends, paid staff, work, organisation, neighbours had an impact on adults with DS in the past and play an important future role [96]. In one study, a six-session programme was designed to help young adults with DS distinguish between friend and boyfriend/girlfriend, effects of jealousy, hurts and trust in relationships, family dynamics, recognising the qualities of a good friend, the impact of gender roles on relationships, and the nature of adult relationships and marriages and found that it helped them to better understand friendship and family life [65]. Another study reported no statistical difference related to gender in adults with DS's self-perception of interpersonal relations [89].

In terms of intimate relationships, many adults with DS had a partner [62, 63, 68, 74, 76, 80, 93], were engaged to their partners [76], wanted to have partner [79, 93] and had a strong desire to marry and have children [97]. Intimate relationships were a source of joy, purpose,

and emotional support [97]. Two studies reported that adults with DS participated in sexual education [80, 98]. One parental-proxy reported study found that adults with DS were independent but did not engage in sexual relationships [80], another study reported that mothers did not consider their adults with DS relationships important [96]. In another study, a mother found a way to support her son with DS who wanted to live with his girlfriend with DS; however, both required assistance with their daily activities such as preparing food, dressing and getting ready for work [83]. Two studies found that adults with DS were subject to a paternalistic care regime by their parents, mostly mothers [79, 80].

## Social inclusion

Nineteen studies examined social inclusion in adults with DS. Findings from five community integration studies showed that adults with DS were active in their communities including volunteering [95], helping people [63, 98] and engaged in social interactions that were described as joyful and positive experiences of being part of a community [93, 97]. One study reported that "institutionalised" adults with DS had a greater sense of community, were more determined to persuade their colleagues and gained more appreciation than those who were cared for by a family [84]. Three studies reported that adults with DS felt accepted and loved by everyone [63, 84, 88]. Male adults with DS had higher levels of social inclusion than females in football teams, and caregivers also showed significant gender-related differences for adults with DS with males being favoured for social inclusion [89].

Other social inclusion indicators reported in adults with DS were discrimination [62], bullying in the community [68, 93], social isolation and social withdrawal [80, 87], withdrawal due to poor communication [93], barriers to participation in health promotion programmes due to lack of energy, boredom, finding them too difficult, and health concerns [92] and appointment reminders via digital assistants to participate in community events [73]. Additionally, amid the COVID-19 pandemic, a significant increase in social withdrawal was observed in the post-lockdown period, affecting the functional and psychosocial well-being of adults with DS [99]. In eight studies, formal/informal support was provided to adults with DS [63, 83, 88, 93, 95–98]. Three studies highlighted the concerns of proxies about adults with DS such as the inadequacy of social networks and their dissatisfaction with individualised support [98], a lack of integration into society [100], a lack of participation in social groups, inadequate healthcare needs (Medicaid), and a desire for acceptance in mainstream schools and the workplace [74].

## Rights

Ten studies examined rights in adults with DS. Three self-reported studies found that adults with DS valued their rights and privacy and wanted to be treated with the same respect and equality as non-disabled people [62, 63, 68]. Likewise, two proxy reported studies stated that mothers agreed that their adults with DS had the same rights and needs as their non-disabled peers in the right to access sex education programmes, and felt it was their moral duty of care to sometimes act as a proxy decision-maker for their adults with DS [79, 80].

A self-and proxy-reported study found no statistical differences regarding the rights of adults with DS [89]. Two studies reported that some adults with DS displayed self-advocacy, for their peers and created public awareness [95, 97]. One study found that the success of the transition process into adulthood for young adults with DS was associated with the level of strong advocacy by mothers, such as securing day placement or employment option and ensuring they had enough activity to be fully occupied 5 days per week [78] and another study reported parents demonstrated relentless advocacy by initiating the care and services for their

adults with DS [97]. One study reported that caregivers indicated that adults with DS preferred living together because they felt accepted and had their right to choose where to live [74].

## Emotional well-being

Thirty studies focused on emotional well-being, with seven studies reporting on the hopes of adults with DS to achieve their personal goals in areas such as getting married and having children [62, 68, 90, 97], getting a job/new job [62, 64, 68, 98], living independently [62, 64, 68, 90, 95], becoming rich and famous and having a car [68] and learning how to prepare meals [98], as compared to only one self-reported study which found that adults with DS had no future plans [63]. Furthermore, caregivers recognised that adults with DS wanted to get married [74, 76, 98] and get a new job [100]. Seven studies reported that adults with DS were satisfied and had good QoL [62, 64, 68, 78, 95, 97, 98].

One study showed no significant impact of aquatic exercise on their QoL in terms of personal satisfaction [71]. Based on self-reports and proxy reports, male adults with DS had higher levels of emotional well-being than females in terms of personal satisfaction, motivation, being stress-free [89]. One study reported that adults with DS were very close and emotionally connected to their non-family network members, but proxies did not know or assumed that the non-family network members did not play an important supportive role [96]. One self-reported study found that enjoyment was demonstrated in three key areas: *interaction* (e.g., social contact with people, exercising, playing with a pet); *achievement* (e.g., completing a task and receiving material rewards); and *process* (e.g., performing a physical activity) [67].

Seven studies reported significant improvements in self-esteem, self-confidence, and self-acceptance via digital assistant [85], enthusiasm and persistence in learning [62, 63, 68, 95, 100] and learning mathematics [90]. In contrast, two studies reported on low self-esteem in adults with DS [71, 93]. There was a reduction in anxiety among adults with DS in the experimental group compared to the control group, and an average decrease in personal growth and self-acceptance in both groups [73]. Findings from one study reported that those living with families had better self-image and good clothing choices than those living in institutions [84]. Another study examined perceptions and performance of adults with DS over a six-year period, which included emotional needs and social skills development in relation to their QoL and showed a slight improvement in self-image for adults with DS in the intervention group, but a deterioration for the non-intervention DS group [88]. Two studies using Emotional Quotient Inventory: Short Version for DS (EQ-i: SVDS) and Bar-On Emotional Quotient Inventory: Youth Version (EQ-i:YV), found they were useful to evaluate the emotional intelligence of adults with DS [69, 72]. Two studies emphasised that spirituality was a strength for adults with adults with DS and documented a range of consistent spiritual practices such as engaging in prayer, meditation, and church attendance [82, 97]. Other reported strengths include the use of humour and a strong appreciation for beauty and excellence or a sense of awe [97]. Two studies reported that adults with DS were less aggressive and rarely fought [84, 99]. Five studies reported high levels of depression manifesting in the form of loneliness [87], poor attitudes towards exercise [92], negative post-lockdown experiences during the COVID-19 pandemic [99] and living in institutionalised homes [83, 84].

Further indicators of emotional well-being in adults with DS included sadness and difficulty comprehending that their parents would die someday [63, 91], negative experiences with external support [62, 98], experiencing mood swings, crying more often, becoming noisier and experiencing self-harm [84]. Similarly, seven studies reported on parental fears, particularly mothers, for their adults with DS such as getting pregnant, becoming the victim of abuse and exploitation in relationships [79, 80], continuity of care in the event of parents' demise [74, 78,

83, 91, 100], uncertainty and struggles of having a good QoL when adults with DS are older [78, 91].

## Physical well-being

Twenty-five studies evaluated the physical well-being of adults with DS. Four studies examined the weight of adults with DS and found that they wanted to lose weight [64, 74, 76, 98], while another study reported that adults with DS made efforts to live a healthy life by maintaining a balanced diet and trying to get regular exercise [91]. Based on self-reports and proxy reports, male adults with DS had higher levels of physical well-being than females [89]. One study found a significant worsening of mental distress, reduced psychosocial well-being, and functional impairments in adults with DS during the post-COVID-19 lockdown [99] while another study reported the lowest number of mental health conditions and required the least behaviour support needs among adults with DS [70].

Eighteen studies reported adults with DS engaged in leisure activities such as sport activities (exercise), listening to music, and watching television [62–64, 67, 68, 71, 76, 81, 82, 84, 86, 88, 90–92, 95, 98, 100]. Two studies showed that adults with DS frequently participated in social and passive leisure activities with low participation in physically and mentally stimulating leisure activities [82, 86]. Despite participating in regular exercise, adults with DS encountered barriers such as lack of energy, boredom, finding it too difficult, health concerns, and uncertainty as to whether the exercises were beneficial to their bodies [92]. Similarly, adults with DS experienced a decline in leisure activities over six years [88] and were neither physically active nor involved in any form of exercise, instead, they preferred to stay at home to access to television, video games, and iPads [74]. Most leisure activities took place at day placement, home, with their family or with a paid carer due to a lack of friends [78]. Three studies found that family members played an important role in facilitating adults with DS involvement in leisure activities and recreation [82, 83, 86].

One study developed a *Servicios de Asistencia Médica de Urgencias Disability Fitness Battery* (SAMU-DISFIT) and found that it was reliable and feasible to measure flexibility, cardiorespiratory fitness, musculoskeletal fitness, and motor fitness in adults with DS [75]. There was no significant difference in the impact of aquatic exercise on the QoL in adults with DS regarding their health [71] and the level of health care needs was not an intervening condition [83]. A proxy reported study found that caregivers were responsible for medical costs since health benefits (Medicaid) did not cover all medical care, especially after the age of 21 [74].

## Material well-being

A total of nineteen studies focused on the material well-being of adults with DS, and most were employed. For example, eighteen studies found that adults with DS enjoyed their work and described it as essential to a sense of independence, friendship, and community participation, but had low income: jobs included sheltered workshops [68, 81, 95], kitchen porters [62, 91, 98], and other forms of employment [63, 64, 67, 70, 74, 76, 78, 90, 94, 96, 97, 100].

Adults with DS were more employed than other disability groups; however, they worked fewer hours [70] and only 15% used computers at work [94]. One study reported that the main reasons for working were individual interest, capability, and building mastery [64]. Adults with DS living with families had a better career concept compared to those living in institutions [84] with males having a higher material well-being than females [89]. One study reported that little public funding was allocated to adults with DS for welfare [74]. Three studies reported that adults with DS were could not meet their needs because they did not have enough money [62, 67, 68] while two studies found that some do not understand the value of

money [90, 100]. Some reasons for unemployment were job-seeking, being laid off and being sacked [94].

Proxies reported they were largely satisfied that the adults with DS were employed [74]. However, there was competition in the employment sector, resulting in adults with DS struggling to find a job and when they do, they are paid less than those who are not disabled [74, 94], and caregivers feared that their adults with DS would lose benefits if they earned more money by working more hours [94]. Another study reported that mothers had difficulty finding appropriate full- or part-time employment, vocational and day recreation programmes for their adults with DS regardless of how long they had been out of school and common issues were waiting for months and insecurity of employment [78].

## Overall QoL and coverage of the QoL domains

Five studies reported on overall QoL domains among adults with DS [62, 68, 74, 89, 95]. One study found that perception of QoL variables was higher in self-reports compared to proxy reports [89]. Furthermore, the results obtained from adults with DS showed significant differences for all variables (domains), including the QoL index, and no significant differences in any of the aspects evaluated by caregivers in athletes and non-athletes [89]. A self-reported study described good QoL as influenced by most components of the International Classification of Functioning, Disability and Health (ICF) such as environmental factors (e.g., supportive social networks and family relationships) and activity and participation factors (e.g., employment and education opportunities, involvement in recreation and leisure activities) [68].

Findings across all included studies showed that QoL can be understood as a dynamic network of domains, with each domain covering a set of strongly connected QoL indicators. During the data synthesis, it was observed that the indicators that describe the QoL domains partially overlap and are intertwined. For example, adults with DS who were prepared to work in a place provided them with a sense of mastery and the opportunity to participate in a community shows that the 'sense of mastery' is part of 'emotional well-being' and 'personal development'; 'work' (under material well-being) and 'participate in a community' (under social inclusion). As such, the core QoL domains are intertwined, and not constructed as clearly distinct entities.

Table 3 shows the coverage of the QoL domain in each study. It provides an overview of how comprehensively and consistently QoL indicators categorised into these domains have been reported in the studies. In descending order, the QoL domains: emotional well-being, physical well-being, self-determination, personal development, interpersonal relations, material well-being and social inclusion were covered in almost all studies, while the least covered was rights.

## Discussion

This is the first mixed methods systematic review to synthesise the evidence on QoL in adults (aged ≥ 18 years) with DS via self-reports and proxy reports using Schalock and Verdugo's QoL model. Based on empirical studies published over 30 years, this review of 39 peer-reviewed publications identified the evidence on QoL in adults with DS and add five unique contributions discussed in the succeeding sections.

### QoL is multidimensional with intertwined domains

All authors of the included studies reported on areas of QoL core domains, suggesting that QoL is a multidimensional concept. However, findings from the heterogeneous designs using

**Table 3. Coverage of core domains in the included studies.**

| References | QoL Domains | | | | | | | |
| --- | --- | --- | --- | --- | --- | --- | --- | --- |
| | Personal development | Self-determination | Interpersonal relations | Social inclusion | Rights | Emotional well-being | Physical well-being | Material well-being |
| Ailey et al. (2006) [87] | | | ✓ | ✓ | | ✓ | | |
| Alderson (2001) [62] | ✓ | ✓ | ✓ | ✓ | ✓ | ✓ | ✓ | ✓ |
| Allahyari and Wolf-Branigin (2018) [74] | ✓ | ✓ | ✓ | ✓ | ✓ | ✓ | ✓ | ✓ |
| Brown (1994) [88] | ✓ | ✓ | | ✓ | | | ✓ | |
| Brown et al. (2001) [63] | | ✓ | ✓ | ✓ | ✓ | ✓ | ✓ | ✓ |
| Bush and Tasse (2017) [70] | ✓ | ✓ | | | | | ✓ | ✓ |
| Cabeza-Ruiz et al. (2019) [75] | | | | | | | ✓ | |
| Camacho et al. (2021) [89] | ✓ | ✓ | ✓ | ✓ | ✓ | ✓ | ✓ | ✓ |
| Carr (2008) [76] | ✓ | ✓ | ✓ | | | ✓ | ✓ | |
| Collacott (1992) [77] | ✓ | | | | | | | |
| Dyke et al. (2013) [78] | ✓ | ✓ | ✓ | | ✓ | ✓ | ✓ | ✓ |
| Faragher and Brown (2005) [90] | ✓ | ✓ | | | | ✓ | ✓ | ✓ |
| Finkelstein et al. (2020) [91] | | ✓ | ✓ | | | ✓ | ✓ | ✓ |
| Foley (2013) [80] | | ✓ | ✓ | ✓ | ✓ | ✓ | | |
| Foley (2014) [79] | | ✓ | ✓ | | ✓ | ✓ | | |
| Goldstein (1988) [81] | | | ✓ | | | | ✓ | ✓ |
| Heller et al. (2004) [92] | | | | ✓ | | ✓ | ✓ | |
| Jackson et al. (2014) [93] | ✓ | ✓ | ✓ | ✓ | | ✓ | | |
| Jevne et al. (2021) [64] | ✓ | ✓ | ✓ | | | ✓ | ✓ | ✓ |
| Jobling et al. (2000) [65] | ✓ | ✓ | ✓ | | | | | |
| Kumin and Schoenbrodt (2016) [94] | ✓ | | | | | | | ✓ |
| Landuran and N'Kaoua (2021) [85] | ✓ | ✓ | | | | ✓ | | |
| Landuran et al. (2022) [66] | | ✓ | | | | | | |
| Landuran et al. 2022 [73] | ✓ | ✓ | | ✓ | | ✓ | | |
| Li et al. (2006) [95] | ✓ | ✓ | ✓ | ✓ | ✓ | ✓ | ✓ | ✓ |
| Love and Agiovlasitis (2016) [67] | ✓ | | | | | ✓ | ✓ | ✓ |
| Mihaila et al. (2020) [86] | ✓ | | | | | | ✓ | |
| Mihaila et al. (2017) [82] | ✓ | | | | | ✓ | ✓ | |
| Pérez et al. (2018) [71] | | | | | | ✓ | ✓ | |
| Robles-Bello et al. 2022 [69] | | | | | | ✓ | | |
| Roll and Bowers (2019) [83] | | | ✓ | ✓ | | ✓ | ✓ | |
| Roll and Koehly (2020) [96] | | ✓ | ✓ | ✓ | | ✓ | | ✓ |
| Sánchez-Teruel et al. (2020) [72] | | | | | | ✓ | | |
| Schroeder-Kurth et al. (1990) [84] | ✓ | ✓ | | ✓ | | ✓ | ✓ | ✓ |
| Scott et al. (2014) [68] | ✓ | ✓ | ✓ | ✓ | ✓ | ✓ | ✓ | ✓ |
| Thompson et al. (2020) [97] | ✓ | ✓ | ✓ | ✓ | ✓ | ✓ | | ✓ |

*(Continued)*

**Table 3.** (Continued)

| References | QoL Domains | | | | | | | |
|---|---|---|---|---|---|---|---|---|
| | Personal development | Self-determination | Interpersonal relations | Social inclusion | Rights | Emotional well-being | Physical well-being | Material well-being |
| Thomson et al. (1995) [100] | ✓ | ✓ | | ✓ | | ✓ | ✓ | ✓ |
| van Heumen and Schippers (2016) [98] | ✓ | ✓ | ✓ | ✓ | | ✓ | ✓ | ✓ |
| Villani et al. (2020) [99] | ✓ | | | ✓ | | ✓ | ✓ | |
| **Total** | 24/39 | 25/39 | 20/39 | 19/39 | 10/39 | 30/39 | 25/39 | 19/39 |

✓ = reported in the study.

the convergent integrated synthesis approach explained QoL as a dynamic network of domains that are intertwined and partially overlapping. The results are similar to van Leeuwen and colleagues [102], who noted that QoL domains are intertwined, when something occurs in one domain, it affects the rest of the network. Considering the WHO definition of QoL, some of the studies reviewed, only partly assessed QoL. For example, the least-covered domain was rights indicating the lack of and need for further empirical studies. The United Nations Convention on the Rights of Persons with Disabilities (UNCRPD) strongly argues that all people with disabilities must enjoy all human rights and fundamental freedoms [103]. The most extensive QoL domain covered across the studies was emotional well-being ($n = 30$), followed by physical well-being ($n = 25$), self-determination ($n = 25$), personal development ($n = 24$), interpersonal relations ($n = 20$), material well-being ($n = 19$), social inclusion ($n = 19$) and rights ($n = 10$).

## A summary of findings of the eight core QoL domains

This review found that the strongest evidence was on *emotional well-being* compared to other domains and suggested that adults with DS expressed how they feel, and how they deal with life events, emphasising their dreams and desires for the future and their *right* to good things in life. The finding on emotional well-being contrasts a scoping review of QoL in children with DS and family variables which revealed that little attention was paid to the emotional well-being of children [47]. A key aspect of QoL is *self-development*; however, this is hampered for some adults with DS due to challenges with acquiring and managing numeracy skills, resulting in difficulty with shopping and handling money. Adults with DS achieved numeracy skills when they were taught within the context of their daily circumstances supports the work of Faragher [104] who confirmed that people with DS learn numeracy concepts best through good teaching and continuous practice and emphasises that numeracy skills should begin in early childhood, continue in schools, and have relevant modifications throughout their adulthood [104]. Numeracy skills are integral to people with DS being able to work and this review identified that many individuals were employed; however, they were often on low income. Prior studies have noted that the importance of higher educational attainment has been associated with better employment outcomes for adults with IDD [105, 106]. Therefore, it is advantageous to encourage young adults with IDD to pursue further education to help improve their chance of employment and income [107, 108]. Participation in employment helps adults with IDD to feel appreciated and may lead to an improvement in self-identity [109]. Under the *material well-being domain*, this review identified some adults with DS who were employed (paid and unpaid) and derived pleasure in helping people.

In the domain of *interpersonal relations*, most adults with DS were dating and planning to get married while others desired friendships, intimate relationships and becoming a parent, yet caregivers were concerned if adults with DS understood what is involved in having relationships and expressed their own fears. A study with people with IDD revealed that 85% wanted a romantic relationship whereas only 35% were in one [110] linking to societal perceptions of disabled people being perceived as 'asexual' [111]. Research shows that young people with DS face challenges when participating in social roles such as having relationships and community compared to daily activities [112]. Another study found that the QoL of adolescents with DS are negatively linked to a lack of friends and health problems [9]. A strong finding of this current review was the invaluable support from family members to their adults with DS which was essential to their QoL. For *physical well-being*, most adults with DS were engaged in passive and active leisure activities and some had poor eating habits which led to poor health. One of the issues that emerged from some of the included studies was the interference with the freedom of adults with DS in the form of paternalism. Therefore, it is worth considering focusing more on self-reported QoL data than proxy data, as this would encourage more independence (under *self-determination*) in adults with DS. Although self-reports and proxy reports are necessary, steps should be taken to enhance concordance. Some studies also revealed that adults with DS encountered bullying, restricted independence, and partial integration in their community (under *social inclusion*). Disabled people have been found to have experiences such as verbal abuse, harassment, and sexual abuse that impact their QoL [113]. Only one study [99] investigated the impact of COVID-19 on adults with DS, but evidence suggested a significant decline in QoL, inferring the need for more studies in this aspect.

## Low reports of good QoL in adults with DS

Of the 39 studies included in this review, only seven studies clearly reported that adults with DS have "good" QoL, which was described based on different aspects of the ICF. The studies presented so far demonstrate that adults with DS rated their QoL higher than proxy reported QoL, showing a discrepancy between the two types of reporting. A possible explanation for the difference could be because adults with DS may have little to compare due to the limited range of experiences and being content with their QoL and may not appreciate the world as much as their caregivers. Caregivers may have compared the QoL in adults with DS to their own personal QoL because they have a much broader range of experience and therefore rate the QoL of adults with DS as low. In accordance with the present results, previous studies have shown inconsistencies in QoL assessment report types [114, 115]. Furthermore, it is difficult to conclude whether this review contradicts earlier empirical research by Albrecht and Devlieger [116], pioneers of the disability paradox, who revealed that most disabled people have good or excellent QoL which contrasts with perceptions of nondisabled people who feel disabled people tend to live an unfavourable daily existence. To date, there is limited information on what represents a "good" QoL for people with IDD [117]. Therefore, it is imperative to carry out further research involving the collation and analysis of DS statistics to ascertain the level of QoL of adults with DS which will also be advantageous in enabling governments to develop and implement strategies specifically targeted to improve their QoL. The absence of such evidence could hinder the individuals from achieving their desired outcome. In this review, proxy reports (mainly from mothers) emphasised the uncertainty of adults with DS attaining a "good" QoL when the adults are older.

## Inconsistency in the use of QoL instruments

The evidence presented in the included studies is inconsistent because there were variations in the instruments used to measure or assess the QoL in adults with DS, as 47 were generic and

test batteries and 4 were DS-specific QoL instruments, making a total of 51 instruments. For the DS-specific QoL instruments, two scales were modified and the number of items was not reported [71, 89], the psychometric properties of two scales were satisfactory [72, 89] and two measured the emotional well-being QoL domain [69, 72]. In the QoL assessment across the studies, a greater proportion of reporting methods was a combination of self- and proxy-report (41.02%) followed by proxy-report (28.20%), self-report (20.51%) and, self-report with minimal support from proxies (10.25%). Greater participation with adults with DS could be facilitated by using cognitive assistive devices, which in turn promotes independence and inclusivity in society [118]. Interestingly, three studies reported the usefulness of ICT in the QoL in adults with DS [64, 73, 85], thus, encouraging self-determination and improvement in other facets of their QoL.

## Similarities and differences in self-reports and proxy reports QoL

The main agreement between self-reported and proxy reported QoL was on poor integration in their community which affected the QoL in adults with DS (in the *social inclusion domain*); although some adults with DS report feeling appreciated in their community. Studies have shown that adults with IDD experience segregation in their community and frequently feel bored [119]. Probable barriers on social inclusion of adults with IDD are experiences of negative attitudes, lack of digital literacy skills and lack of supportive social network which are detrimental to their QoL [120].

The major disparity in the self-and proxy reported QoL was on independence (in the *self-determination domain*); this was common among younger adults who felt their independence was hindered by their caregivers and relationships (*interpersonal relations domain)*. The *telos* of adults with DS might be different from the caregivers. The subjective satisfaction of adults with DS may differ from the objective assessment of an independent third party (e.g., their family members or other caregivers) and this may therefore explain a gap between the self-report and the proxy report on QoL. The differences would then be due to subjectivity versus objectivity, and different aims and/or values of the reporters. This review finding supports Heaslip and Hewitt-Taylor [121], who noted that reducing opportunities for vulnerable people to take risks encourages vulnerability. People with IDD desire to live independently [122, 123] and want to exercise their human rights—a key facet of autonomy, via supported decision-making with the assistance of their caregivers and promoted by the ability to communicate their decisions to others [124]. In this review, caregivers (particularly mothers) had concerns about the level and kind of support their adults would receive in the future.

## Strengths and limitations of the study

Notably, a major strength of the review was its conceptual foundation in a well-recognised framework: Schalock and Verdugo QoL model [58]. The framework provided a useful structure that contributed to the identification of the extent to which eight core dimensions of QoL have been examined. This review employed a comprehensive search strategy to capture most of the evidence. Another strength is the approach to synthesise and integrate all quantitative, qualitative and mixed methods designs using the convergent integrated approach. However, several limitations of this study should be noted. First, search strategies using databases that privilege certain types of journals might limit the inclusion of LMICs, and the impact of articles published only in English might have excluded some valuable studies. Second, all included studies were conducted in HICs, which may have introduced bias and limited the generalisability of the findings. In most cultures, IDD is accompanied by marginalisation and stigmatisation [125, 126]. It would be interesting to examine what extent the QoL of adults with DS via

self-and proxy reports is universally bound. Evidence shows that culture plays a vital role in shaping an individual's QoL [127]. An individual's values might have an impact on their QoL, and this can differ between cultures [127]. Third, a single reviewer performed the screening, data extraction and quality assessment, which may have introduced bias, although a random sample of potentially eligible full-text articles was independently cross-checked by the entire team.

Finally, methodological issues based on the heterogeneity of the extracted data did not allow statistical analysis (meta-analysis). The differences associated with the methodological designs of the included studies might affect the synthesis of the findings. However, a standard-ised synthesis approach and quality appraisal tool were used to integrate the quantitative and qualitative evidence and assess the quality of included studies that informed the discussion of findings. Yet, the findings of the review need to be interpreted with caution. Most of the included studies used cross-sectional designs and qualitative approaches that make it difficult to identify changes in QoL requirements over time to provide effective strategies to improve QoL in this population. Further research on QoL among adults with DS should place greater emphasis on incorporating longitudinal studies that address the predictors of QoL in adults with DS over time due to transition from school to adult life, young adults to older adults, and the desire of adults to exercise their rights to live independently. Several researchers have sug-gested more longitudinal studies from a well-designed population-based registry as it would allow investigators to examine the relationship in the QoL domains and indicators alongside changes that might have occurred over time [33, 76, 98, 128].

## Conclusions

This review has demonstrated preliminary evidence on QoL among adults with DS. The results highlight gaps in the current body of research and promising areas for conducting more stud-ies. Future research should take into consideration flexible QoL instruments that are culture-specific and focused on adults with DS for a complete assessment of QoL. Findings showed the significant QoL needs among adults with DS including fulfilling their desire for full indepen-dence, have relationships, involvement in community participation and exercise their human rights. Overall, the review shows persistent inconsistencies in evidence and perhaps this is a consequence of data collection methods and QoL reporting methods. Further studies need to investigate the differences between self-reported and proxy reported QoL in this population. This review encourages the use of innovative technology tools such as the Smart Angel system, which uses a cloud-based support and monitoring system [129, 130] to obtain self-reported QoL data in adults with DS, as this would encourage inclusion in society.

## Supporting information

**S1 File. PRISMA 2020 checklist.**
(DOCX)

**S1 Table. Search strategy.**
(DOCX)

**S2 Table. Summary characteristics of the included studies.**
(DOCX)

**S3 Table. Quality assessment of included studies (quality appraisal).**
(DOCX)

## Acknowledgments

We thank Dr Sid Carter for his valuable contribution in the development of the review protocol alongside Mr José López Blanco and Mr Caspian Dugdale for their contribution on the search strategies.

## Author Contributions

**Conceptualization:** Ogochukwu Ann Ijezie, Vanessa Heaslip.

**Data curation:** Ogochukwu Ann Ijezie.

**Formal analysis:** Ogochukwu Ann Ijezie.

**Methodology:** Ogochukwu Ann Ijezie, Jane Healy, Philip Davies, Emili Balaguer-Ballester, Vanessa Heaslip.

**Project administration:** Ogochukwu Ann Ijezie.

**Supervision:** Jane Healy, Philip Davies, Emili Balaguer-Ballester, Vanessa Heaslip.

**Visualization:** Ogochukwu Ann Ijezie.

**Writing – original draft:** Ogochukwu Ann Ijezie.

**Writing – review & editing:** Ogochukwu Ann Ijezie, Jane Healy, Philip Davies, Emili Balaguer-Ballester, Vanessa Heaslip.

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
