## [Decision Letter · Decision Letter 0]

10 Oct 2022

PONE-D-22-24936Quality of life in adults with Down syndrome: A mixed methods systematic reviewPLOS ONE

Dear,

Thank you for submitting your manuscript to PLOS ONE. After careful consideration, we feel that it has merit but does not fully meet PLOS ONE’s publication criteria as it currently stands. Therefore, we invite you to submit a revised version of the manuscript that addresses the points raised during the review process. Please submit your revised manuscript by 24 November 2022 If you will need more time than this to complete your revisions, please reply to this message or contact the journal office at plosone@plos.org. Please include the following items when submitting your revised manuscript:A rebuttal letter that responds to each point raised by the academic editor and reviewer(s). You should upload this letter as a separate file labeled 'Response to Reviewers'.A marked-up copy of your manuscript that highlights changes made to the original version. You should upload this as a separate file labeled 'Revised Manuscript with Track Changes'.An unmarked version of your revised paper without tracked changes. You should upload this as a separate file labeled 'Manuscript'.

We look forward to receiving your revised manuscript.

Kind regards,

Muhammad Shahzad Aslam, Ph.D.,M.Phil., Pharm-D

Academic Editor

PLOS ONE

“The authors received no specific funding for this work and was conducted as part of a PhD project.”

Reviewers' comments:

Reviewer's Responses to Questions

**Comments to the Author**

1. Is the manuscript technically sound, and do the data support the conclusions?

Reviewer #1: Yes

Reviewer #2: Yes

2. Has the statistical analysis been performed appropriately and rigorously? 

Reviewer #1: Yes

Reviewer #2: Yes

3. Have the authors made all data underlying the findings in their manuscript fully available?

Reviewer #1: Yes

Reviewer #2: Yes

4. Is the manuscript presented in an intelligible fashion and written in standard English?

Reviewer #1: Yes

Reviewer #2: Yes

5. Review Comments to the Author

Reviewer #1: The abstract of the study presents numerous abbreviations, when the publisher's specifications suggest that acronyms should not appear in the abstract.

I only find this detail. The rest seems impeccable to me.

Reviewer #2: The authors present a good research paper.

• The relevance of the topic: Good.

• Introduction: Good.

• Methodology: Can be improved.

• Results: Good.

• Discussion: Good.

However, ACCEPT AFTER MINOR REVISION. In general, the paper follows an adequate structure and correct scientific support and can be published considering some limitations. The study is interesting in the field of people with Down syndrome. However, there are a series of limitations that should be considered.

In the first place, carry out a review of the existing literature related to the subject, being essential to inquire into the Plos One journal itself, since there are papers related to its manuscript that can help to improve it. Therefore, include those references, if any, especially from the last five years. In addition, recommend reading some papers related to the topic of adults with down syndrome:

Gámez-Calvo, L., Gamonales, J. M., León, K., & Muñoz-Jiménez, J. (2022). Influence of Balance on the Quality of Life of People with Down Syndrome in School and Adult Ages: A Literature Review. MHSalud, 19(1), 1-15. https://doi.org/10.15359/mhs.19-1.6

Gámez-Calvo, L., Gamonales, J.M., Silva-Ortíz, A.M., & Muñoz-Jiménez, J. (2022). Benefits of hippotherapy in elderly people: Scoping review. Journal of Human Sport and Exercise, 17(2), 302-313. https://doi.org/10.14198/jhse.2022.172.06

Hamadelseed, O., Elkhidir, I. H., & Skutella, T. (2022). Psychosocial Risk Factors for Alzheimer’s Disease in Patients with Down Syndrome and Their Association with Brain Changes: A Narrative Review. Neurology and Therapy, 1-23.

Rodríguez-Grande, E. I., Buitrago-López, A., Torres-Narváez, M. R., Serrano-Villar, Y., Verdugo-Paiva, F., & Ávila, C. (2022). Therapeutic exercise to improve motor function among children with Down Syndrome aged 0 to 3 years: a systematic literature review and meta analysis. Scientific reports, 12(1), 1-11.

Silva-Ortiz, A., Gamonales, J., Gámez-Calvo, L., & Muñoz-Jiménez, J. (2020). Beneficios de la actividad física inclusiva para personas con síndrome de Down: revisión sistemática. SPORT TK: Revista EuroAmericana de Ciencias del Deporte, 9(2), 81–94. https://doi.org/10.6018/sportk.454201

Specific comments.

Title: It is correct.

Abstract. Incorporate in the summary, a more precise sentence of the results.

Introduction. This section presents the problem in a coherent and clear manner with the correct support of the scientific literature. However, it is convenient to update the references, since there are different works related to the subject and no mention is made, and it would even be interesting to mention the different existing works related to technological advances in people with down syndrome. Also, it could be a future study of review.

Methods. Modify the method section and incorporate the sections: Design.

- Study design. To write the design section, we recommend that you take some of the following methodologists as references.

Ato, M., López-García, J. J., & Benavente, A. (2013). A classification system for research designs in psychology. Anales de Psicología/Annals of Psychology, 29(3), 1038-1059.

Montero, I., & León, O.G. (2007). A guide for naming research studies in psychology. International Journal of Clinical and Health Psychology, 7(3), 847-862.

Results. Summary of study data and table are correct.

Conclusion. Differentiate the discussion of the main conclusions of the work. To do this, you must create this section. And modify the limitations of the study and locate them in said section at the end. Also, they must be direct, and highlight the main contributions of the study.

References. They should be reviewed and updated according to the publication standards.

6. PLOS authors have the option to publish the peer review history of their article (what does this mean?). If published, this will include your full peer review and any attached files.

Reviewer #1: No

Reviewer #2: No

---

## [Author Response · Author response to Decision Letter 0]

9 Dec 2022

Thank you for the email dated 10 October 2022. 

We are delighted that both you and the reviewers highlighted that the paper follows an appropriate structure, is properly scientifically grounded, and is of interest in the field of people with Down syndrome. We would like to thank you and the reviewers for your suggestions to further strengthen the paper and for discovering relevant ideas for future research.

We would like to express our sincere gratitude for taking your precious time to provide in-depth comments that have significantly improved the quality of our paper. We have reviewed the comments and generally agree with them. Where possible, we have made changes to the manuscript based on your observations and those of the reviewers. 

We have discovered 2 recently published articles and added them to our review. Instead of 37 studies initially included, we have 39 studies. For the sake of transparency, we have marked the following in green: i) The new included papers in the S2 Table and S3 Table. ii) Updated page numbers in S1 Checklist PRISMA 2020 Checklist. We have updated Fig 1 and Fig 2. We also marked your comments (journal requirements) and those of the reviewers in black and our answers are marked as the “Authors’ Response” in green.

As requested, we have also included the following in our submission:

1. A marked-up copy of our manuscript that highlights changes made to the original version labelled ‘Revised Manuscript with Track Changes.’

2. An unmarked version of our revised paper without tracked changes as a separate file labelled ‘Manuscript.’

We hope that these revisions will give you and the reviewers’ comments sufficient consideration and that the paper will be accepted for publication in the PLOS ONE Journal.

We take this opportunity to thank everyone involved in the process.

Yours sincerely,

Ogochukwu Ann Ijezie 

PhD student

Department of Computing and Informatics

P256, Poole House, Talbot Campus, BH12 5BB

Bournemouth University, United Kingdom

oijezie@bournemouth.ac.uk

Corresponding Author

(On the behalf of all authors)

Authors’ Response: Thank you for your comment. This has been completed.

• Thank you for stating the following financial disclosure:

“The authors received no specific funding for this work and was conducted as part of a PhD project.”

Authors’ Response: We are grateful for your feedback. We chose the fourth option from your comments relevant to our work as it is not funded and also updated the statement in our cover letter. Hence, we have amended our manuscript as follows: “The authors received no specific funding for this work.”

• Please include captions for your Supporting Information files at the end of your manuscript, and update any in-text citations to match accordingly. Please see our Supporting Information guidelines for more information: http://journals.plos.org/plosone/s/supporting-information.

Authors’ Response: Thank you for your comment. This has been completed.

Reviewers' comments:

Reviewer's Responses to Questions

Comments to the Author

Reviewer #1 and Reviewer #2

1. Is the manuscript technically sound, and do the data support the conclusions? Yes

Authors’ Response: We are grateful for your feedback.

2. Has the statistical analysis been performed appropriately and rigorously? Yes

Authors’ Response: We are grateful for your feedback.

3. Have the authors made all data underlying the findings in their manuscript fully available? Yes

Authors’ Response: We are grateful for your feedback.

4. Is the manuscript presented in an intelligible fashion and written in standard English? Yes

Authors’ Response: We are grateful for your feedback.

5. Review Comments to the Author

Reviewer #1: The abstract of the study presents numerous abbreviations, when the publisher's specifications suggest that acronyms should not appear in the abstract.

I only find this detail. The rest seems impeccable to me.

Authors’ Response: Many thanks for raising this point. In response to this comment, we have addressed this by eliminating abbreviations where possible in the abstract (see page 2 of the revised manuscript).

Reviewer #2: The authors present a good research paper.

• The relevance of the topic: Good.

• Introduction: Good.

• Methodology: Can be improved.

• Results: Good.

• Discussion: Good.

However, ACCEPT AFTER MINOR REVISION. In general, the paper follows an adequate structure and correct scientific support and can be published considering some limitations. The study is interesting in the field of people with Down syndrome. However, there are a series of limitations that should be considered.

In the first place, carry out a review of the existing literature related to the subject, being essential to inquire into the Plos One journal itself, since there are papers related to its manuscript that can help to improve it. Therefore, include those references, if any, especially from the last five years. In addition, recommend reading some papers related to the topic of adults with down syndrome:

Gámez-Calvo, L., Gamonales, J. M., León, K., & Muñoz-Jiménez, J. (2022). Influence of Balance on the Quality of Life of People with Down Syndrome in School and Adult Ages: A Literature Review. MHSalud, 19(1), 1-15. https://doi.org/10.15359/mhs.19-1.6.

Gámez-Calvo, L., Gamonales, J.M., Silva-Ortíz, A.M., & Muñoz-Jiménez, J. (2022). Benefits of hippotherapy in elderly people: Scoping review. Journal of Human Sport and Exercise, 17(2), 302-313. https://doi.org/10.14198/jhse.2022.172.06.

Hamadelseed, O., Elkhidir, I. H., & Skutella, T. (2022). Psychosocial Risk Factors for Alzheimer’s Disease in Patients with Down Syndrome and Their Association with Brain Changes: A Narrative Review. Neurology and Therapy, 1-23.

Rodríguez-Grande, E. I., Buitrago-López, A., Torres-Narváez, M. R., Serrano-Villar, Y., Verdugo-Paiva, F., & Ávila, C. (2022). Therapeutic exercise to improve motor function among children with Down Syndrome aged 0 to 3 years: a systematic literature review and meta analysis. Scientific reports, 12(1), 1-11.

Silva-Ortiz, A., Gamonales, J., Gámez-Calvo, L., & Muñoz-Jiménez, J. (2020). Beneficios de la actividad física inclusiva para personas con síndrome de Down: revisión sistemática. SPORT TK: Revista EuroAmericana de Ciencias del Deporte, 9(2), 81–94. https://doi.org/10.6018/sportk.454201.

Authors’ Response: We are thankful for these comments and for raising interesting points. We have revised the methodology section (see pages 5 to 7 of the revised manuscript). We really appreciate the recommended papers. We read them and realised that only paper III was appropriate for our study. For papers I and V, only the abstract was written in English, paper II (it was not suitable for our study as it talked about hippotherapy) and paper IV (it focused on children with Down syndrome aged 0 to 3 years.

As requested, we conducted a search from the last five years (2017 – 2022) for existing literature in the PLOS ONE journal. We found four articles published within the timeline related to our manuscript and added them to the introduction on page 3:

I. Lancioni GE, Singh NN, O’Reilly MF, Sigafoos J, Alberti G, Del Gaudio V, et al. (2022) People with intellectual and sensory disabilities can independently start and perform functional daily activities with the support of simple technology. PLoS ONE 17(6): e0269793. https://doi.org/10.1371/journal.pone.0269793.

II. De-Rosende-Celeiro I, Torres G, Seoane-Bouzas M, Ávila A (2019) Exploring the use of assistive products to promote functional independence in self-care activities in the bathroom. PLoS ONE 14(4): e0215002. https://doi.org/10.1371/journal.pone.0215002.

III. Haddad F, Bourke J, Wong K, Leonard H (2018) An investigation of the determinants of quality of life in adolescents and young adults with Down syndrome. PLoS ONE 13(6): e0197394. https://doi.org/10.1371/journal.pone.0197394.

The aim of paper III was to investigate determinants of health-related quality of life of young people with Down syndrome, with a focus on the identification of risk and protective factors pertaining not only to the individuals themselves but also to their family environment. We did not include it as part of the included studies because our manuscript only focused on the quality of life in adults with Down syndrome. We extracted only relevant information and included it in the introduction (see page 3) and discussion (see page 23) sections.

IV. Gandy KC, Castillo HA, Ouellette L, Castillo J, Lupo PJ, Jacola LM, et al. (2020) The relationship between chronic health conditions and cognitive deficits in children, adolescents, and young adults with down syndrome: A systematic review. PLoS ONE 15(9): e0239040. https://doi.org/10.1371/journal.pone.0239040.

Additionally, we found an article published in 2014 and included it in the discussion section:

V. Foley K-R, Girdler S, Bourke J, Jacoby P, Llewellyn G, Einfeld S, et al. (2014) Influence of the Environment on Participation in Social Roles for Young Adults with Down Syndrome. PLoS ONE 9(9): e108413.https://doi.org/10.1371/journal.pone.0108413.

For paper V, the age group of young adults were: 16 – 18 years (first age group), 19 – 22 years and 23 – 31 years. During analysis, they merged all the age groups together. We did not include it as part of the included studies because our manuscript only focused on aged 18 years and above. We have extracted only relevant information and included it in the discussion section (see page 23 of the revised manuscript).

We hope we addressed this section in line with your request.

Specific comments.

6. Title: It is correct.

Authors’ Response: We appreciate your comment.

7. Abstract. Incorporate in the summary, a more precise sentence of the results.

Authors’ Response: Many thanks for raising this point. In response to this comment, we have addressed this by writing the abstract in a more precise format (see page 2 of the revised manuscript).

8. Introduction. This section presents the problem in a coherent and clear manner with the correct support of the scientific literature. However, it is convenient to update the references, since there are different works related to the subject and no mention is made, and it would even be interesting to mention the different existing works related to technological advances in people with down syndrome. Also, it could be a future study of review.

Authors’ Response: Many thanks for the kind words and for providing excellent suggestions as they helped strengthened the quality of our work, unearthed good ideas and gave a different perspective on proffering solutions for adults with Down syndrome via information and communication technology. We have addressed this by updating the references based on existing works related to the technological advances in people with Down syndrome (see page 3 of the revised manuscript). 

9. Methods. Modify the method section and incorporate the sections: Design.

- Study design. To write the design section, we recommend that you take some of the following methodologists as references.

Ato, M., López-García, J. J., & Benavente, A. (2013). A classification system for research designs in psychology. Anales de Psicología/Annals of Psychology, 29(3), 1038-1059.

Montero, I., & León, O.G. (2007). A guide for naming research studies in psychology. International Journal of Clinical and Health Psychology, 7(3), 847-862.

Authors’ Response: Many thanks for the recommended papers. We read the papers and realised that they were not really suitable for our methods section as they did not specify methodologies used in systematic reviews. Since our study is on mixed methods systematic review, we found two recently published articles to address your comment which are:

I. Stern, C., Lizarondo, L., Carrier, J., Godfrey, C., Rieger, K., Salmond, S., Apostolo, J., Kirkpatrick, P. and Loveday, H., 2020. Methodological guidance for the conduct of mixed methods systematic reviews. JBI evidence synthesis, 18(10), pp.2108-2118. https://doi.org/10.11124/jbisrir-d-19-00169.

II. Lizarondo, L., Stern, C., Apostolo, J., Carrier, J., de Borges, K., Godfrey, C., Kirkpatrick, P., Pollock, D., Rieger, K., Salmond, S. and Vandyk, A., 2022. Five common pitfalls in mixed methods systematic reviews: lessons learned. Journal of Clinical Epidemiology, 148, pp. 178-183. https://doi.org/10.1016/j.jclinepi.2022.03.014

The articles are located on page 5 of the revised manuscript.

10. Results. Summary of study data and table are correct.

Authors’ Response: We are grateful for your feedback in this section. We discovered 2 recently published articles. We extracted the data, appraised the studies and synthesised and added them to our review:

I. Landuran, Audrey, Hélène Sauzéon, Charles Consel, and Bernard N’Kaoua (2022). Evaluation of a smart home platform for adults with Down syndrome. Assistive Technology, pp.1-11. https://doi.org/10.1080/10400435.2022.2075487

II. María Auxiliadora Robles-Bello, David Sánchez-Teruel, Nieves Valencia Naranjo & Rafael Delgado Rodríguez (2022). Preliminary Study on Emotional Competence in Adults with Down Syndrome, International Journal of Disability, Development and Education, 69:6, 2136-2154. https://doi.org/10.1080/1034912X.2020.1840532

There was a numerical error in the manuscript with track changes (see page 15), which has been corrected in the revised manuscript and does not affect the overall result.

11. Conclusion. Differentiate the discussion of the main conclusions of the work. To do this, you must create this section. And modify the limitations of the study and locate them in said section at the end. Also, they must be direct, and highlight the main contributions of the study.

Authors’ Response: Thank you so much for your recommendation. We have differentiated the main findings of the study, moved the limitations at the end and made the conclusions more explicit. We are unsure if we have adequately addressed this as you requested. If not, we are happy to revise it again.

12. References. They should be reviewed and updated according to the publication standards.

Authors’ Response: We have amended and updated our reference list to meet PLOS ONE guidelines.

---

## [Decision Letter · Decision Letter 1]

5 Apr 2023

Quality of life in adults with Down syndrome: A mixed methods systematic review

PONE-D-22-24936R1

Dear Dr. Ijezie,

We’re pleased to inform you that your manuscript has been judged scientifically suitable for publication and will be formally accepted for publication once it meets all outstanding technical requirements.

Kind regards,

Muhammad Shahzad Aslam, Ph.D.,M.Phil., Pharm-D

Academic Editor

PLOS ONE

Additional Editor Comments (optional):

Reviewers' comments:

Reviewer's Responses to Questions

**Comments to the Author**

1. If the authors have adequately addressed your comments raised in a previous round of review and you feel that this manuscript is now acceptable for publication, you may indicate that here to bypass the “Comments to the Author” section, enter your conflict of interest statement in the “Confidential to Editor” section, and submit your "Accept" recommendation.

Reviewer #2: All comments have been addressed

2. Is the manuscript technically sound, and do the data support the conclusions?

Reviewer #2: Yes

3. Has the statistical analysis been performed appropriately and rigorously? 

Reviewer #2: Yes

4. Have the authors made all data underlying the findings in their manuscript fully available?

Reviewer #2: Yes

5. Is the manuscript presented in an intelligible fashion and written in standard English?

Reviewer #2: Yes

6. Review Comments to the Author

Reviewer #2: (No Response)

7. PLOS authors have the option to publish the peer review history of their article (what does this mean?). If published, this will include your full peer review and any attached files.

Reviewer #2: **Yes: **José M. Gamonales

---

## [Editor Report · Acceptance letter]

14 Apr 2023

PONE-D-22-24936R1 

Quality of life in adults with Down syndrome: A mixed methods systematic review 

Dear Dr. Ijezie:

I'm pleased to inform you that your manuscript has been deemed suitable for publication in PLOS ONE. Congratulations! Your manuscript is now with our production department. 

Kind regards, 

on behalf of

Dr. Muhammad Shahzad Aslam 

Academic Editor

PLOS ONE